# Comparison of Different Animal Models in Hindlimb Functional Recovery after Acute Limb Ischemia-Reperfusion Injury

**DOI:** 10.3390/biomedicines12092079

**Published:** 2024-09-12

**Authors:** Nadezhda N. Zheleznova, Claire Sun, Nakul Patel, Nathan Hall, Kristof M. Williams, Jie Zhang, Jin Wei, Lusha Xiang, Ridham Patel, Sahil Soni, Divya Sheth, Enyin Lai, Xingyu Qiu, Nohely Hernandez Soto, Ruisheng Liu

**Affiliations:** 1Department of Molecular Pharmacology and Physiology, University of South Florida College of Medicine, Tampa, FL 33612, USA; zheleznova@usf.edu (N.N.Z.); sunclaire@usf.edu (C.S.); nakul@usf.edu (N.P.); nathanhall@usf.edu (N.H.); williamskristof@usf.edu (K.M.W.); ridhampatel@usf.edu (R.P.); sahilsoni@usf.edu (S.S.); divyasheth@usf.edu (D.S.); nohelyh@usf.edu (N.H.S.); 2Division of Nephrology at Boston Medical Center, Department of Medicine, Boston University Chobanian and Avedisian School of Medicine, Boston, MA 02118, USA; jzhang7@bu.edu (J.Z.); jwei7@bu.edu (J.W.); 3United States Army Institute of Surgical Research, 3698 Chambers Pass BLDG 3611, Ft. Sam Houston, TX 78234, USA; lusha.xiang.civ@health.mil; 4Department of Physiology, Zhejiang University School of Medicine, Hangzhou 310058, China; laienyin@zju.edu.cn (E.L.); qiuxingyu960228@zju.edu.cn (X.Q.)

**Keywords:** limb ischemia, SD rat, pneumatic cuff, ligation, aorta, iliac, femoral, artery, grip strength, motor function, creatine kinase, muscle injury

## Abstract

Acute limb ischemia (ALI) is a sudden lack of blood flow to a limb, primarily caused by arterial embolism and thrombosis. Various experimental animal models, including non-invasive and invasive methods, have been developed and successfully used to induce limb ischemia-reperfusion injuries (L-IRI). However, there is no consensus on the methodologies used in animal models for L-IRI, particularly regarding the assessment of functional recovery. The present study aims to compare different approaches that induce L-IRI and determine the optimal animal model to study functional limb recovery. In this study, we applied a pneumatic cuff as a non-invasive method and ligated the aorta, iliac, or femoral artery as invasive methods to induce L-IRI. We have measured grip strength, motor function, creatine kinase level, inflammatory markers such as nuclear factor NF-κB, interleukin-6 (IL-6), hypoxia markers such as hypoxia-induced factor-1α (HIF-1α), and evaluated the muscle injury with hematoxylin and eosin (H&E) staining in Sprague Dawley rats after inducing L-IRI. The pneumatic pressure cuff method significantly decreased the muscle strength of the rats, causing the loss of ability to hold the grid and inducing significant limb function impairment, while artery ligations did not. We conclude from this study that the tourniquet cuff method could be ideal for studying functional recovery after L-IRI in the rat model.

## 1. Introduction

Acute limb ischemia (ALI) is a sudden decrease in limb arterial perfusion that threatens limb viability and is considered a major vascular emergency. Thrombosis and embolism are common causes of ALI. Emergency revascularization is the gold standard for treating ALI [1]. However, after the restoration of blood flow to the ischemic organ, many patients exhibit further skeletal muscle injuries, compartment syndrome, and multi-organ failures due to the ischemia-reperfusion injury (IRI). Limb IRI (L-IRI) is associated with increased morbidity and disability [1,2,3,4,5,6,7,8].

Various experimental animal models for L-IRI, both invasive and non-invasive, have been utilized to explore pathophysiological mechanisms and identify potential therapeutic targets. Tourniquet-induced L-IRI is the most common method for non-invasive models [9,10]. The invasive models are usually induced by ligating different arteries that supply blood to the lower limbs. Unfortunately, there is lack of consensus on the animal models for L-IRI in methodology. In addition, the number of studies for the long-term functional evaluation of the limbs in the invasive models is limited.

A previous study from the University of Nebraska compared lower L-IRI induced by tourniquet application and infrarenal aortic clamping in rats [11]. The authors found that tourniquet application induced severe lower limb ischemia, while infrarenal aortic ligation did not, primarily due to the rich collateral system of the lower limbs. However, this study was a non-survival experiment with only 4 h of reperfusion and did not assess limb function [11]. L-IRI induced by tourniquet application and ligation of the femoral artery were also compared in mouse models [12]. Another study compared different durations of tourniquet application, ranging from 1 to 3 h, that induced L-IRI. They found that muscle damage measured with enzyme levels and histology was correspondingly enhanced with the extension of the tourniquet duration [13]. A similar study compared different tourniquet durations, ranging from 1 to 6 h, in a mouse model. They found corresponding alterations in blood flow and histological changes with extension of tourniquet duration. However, this study only observed 24 h of reperfusion and did not evaluate the limb function [14].

Our present study compares several commonly used L-IRI models in long-term functional recovery. We used a pneumatic cuff as a non-invasive method and ligated the aorta, iliac, or femoral artery as an invasive method. We observed impaired limb functional recovery in the group subjected to pneumatic-cuff-induced L-IRI, while ligating the arteries did not result in any functional impairment.

## 2. Materials and Methods

### 2.1. Animals

The animal use and welfare procedures adhered to the National Institutes of Health (NIH) Guide for the Care and Use of Laboratory Animals. The Institutional Animal Care and Use Committee (IACUC) of the University of South Florida (USF) College of Medicine approved the study. Male Sprague Dawley (SD) rats, aged 8–10 weeks (250 to 300 g), were obtained from Envigo. After arrival, the animals were housed in the animal facilities at USF, had access to food and water ad libitum, and were under a 12-h:12-h light-dark cycle. The animals were randomly assigned to the sham group, pneumatic cuff, or various ligation groups, depending on the experimental requirements.

### 2.2. Non-Invasive Method: Pneumatic-Cuff-Induced L-IRI

SD rats were anesthetized with pentobarbital (100 mg/kg of body weight). Slow-release buprenorphine (1.2 mg/kg), a pain killer, was injected subcutaneously in the scruff (loose skin on the dorsal neck) using a 27 g needle prior to the experiment. The animals were placed in a ventral position, and fur from the left hindlimb was removed using hair clippers (Oster) and depilatory cream (Veet) for complete hair removal. After this, the left limb was cuffed with a customized disposable vascular cuff (DP2.5™, Hokanson, Bellevue, WA, USA) connected to a pressure-controlled inflator (E20 Rapid Cuff Inflation System, Hokanson, Bellevue, WA, USA). Then, a pressure of 250 mmHg was applied using a cuff inflator air source (AG101, Hokanson). The animal was constantly monitored and put under light isoflurane anesthesia by mask (maintenance: 1% isoflurane and O_2_ 2 L/min) when necessary. The body temperature was maintained continuously throughout the procedure using a heating pad (Temperature Controller, CMA-Harvard Apparatus, Holliston, MA, USA). After 3 h, the cuff was removed, and the animal was put in a single cage for recovery from anesthesia. Sham groups were performed following the same procedures and time course except without the application of cuff pressure.

### 2.3. Invasive Methods: Ligation-of-Arteries-Induced L-IRI

Surgeries were performed under isoflurane anesthesia (2–3% with O_2_ at 2 L/min rate). Buprenorphine SC analgesic (0.1 mL/200 g) was injected subcutaneously approximately 30 min before the completion of the ischemic period. Animals were placed in supine position (lying horizontally with the face up) on a heating pad (Temperature Controller, CMA-Harvard Apparatus, Holliston, MA, USA). The fur was removed from the area of the incision. The surgeries were completed under sterilized conditions. The following surgeries were performed in separate groups of animals: (1) abdominal aorta clamping: an abdominal incision was performed. The abdominal aorta below the renal arteries was carefully dissected and isolated from the surrounding tissues and veins for clamping; (2) iliac artery clamping: an abdominal incision was performed. The left common iliac artery was carefully dissected and isolated from surrounding tissues and veins for clamping; and (3) femoral artery clamping: an incision on the angle of the left hind leg was made. The femoral artery was carefully dissected and isolated from surrounding tissues and veins for clamping. The aorta, femoral, or iliac artery was clamped with a micro-serrefine clamp (#18055-06, FST) for 3 h. Upon completion of a 3 h period of ischemia, the muscle was sutured using a 4-0 absorbable suture (Hexa 4-0, #YG10029-1), and the skin was stapled with Autoclip (MikRon, Biel/Bienne, Switzerland) 9 mm wound clips. The animal was returned to the single cage to recover from anesthesia. Sham groups were performed following the same surgical procedures and time course except without the ligation of the vessels.

### 2.4. Grip Strength Measurements in SD Rats

The grip strength of the hindlimb was measured using a grip strength meter (47200 Ugo Basile Grip Strength Meter, Gemonia VA, Italy) at 24 h and up to 7 days post-L-IRI. This system assessed functional recovery by automatically measuring the grip strength (i.e., peak force and time resistance) of the hindlimb in rats, using the grid for evaluation [15,16,17]. The animal was placed over a base plate, in front of a grasping tool (grid). The bar was mounted to a force sensor connected to the control unit, which was connected to the computer via the USB port for monitoring and data recording using the provided software. An observer who was unaware of the treatment groups performed the measurement and analysis.

### 2.5. Modified Tarlov Scale

A scoring system, the modified Tarlov scale, was used to assess the functional recovery of the experimental hindlimb of the rat 24 h and up to 7 days after L-IRI. The animal’s position, motion against gravity, gait, and walking ability were considered during scoring. Animals were observed in a transparent 1 × 2 ft bucket where their movements were unobstructed; grading of the functional recovery was conducted by a third party unaware of the animal’s category. Scores were compared with the contralateral hindlimb, which is normal, as described in Table 1 [13,18].

### 2.6. Creatine Kinase Assay

Creatine kinase (CK) is widely regarded as the most sensitive marker of muscle injury. In this study, plasma CK levels were measured at 24 h post-ischemia. A rat creatine kinase ELISA (enzyme-linked immunosorbent assay) kit (MBS1600481) was used, featuring a standard curve range of 0.05–30 ng/mL and a sensitivity of 0.04 ng/mL, with 96 wells for quantitative measurement of CK in collected plasma samples. CK catalyzes the conversion of creatine to phosphocreatine using ATP, and the phosphocreatine and ADP produced react with a CK enzyme mix to form an intermediate. This intermediate reduces a colorless probe to a colored product with strong absorbance at λ = 450 nm. Before running the assay, the kit and samples were brought to room temperature for 30 min. The assay plate was pre-coated with a primary antibody against rat CK, and a biotinylated rat CK antibody was used as a secondary antibody. Streptavidin-HRP served as the detection biomarker. Upon addition of a substrate solution, a color change proportional to the quantity of rat CK in the sample developed. The reaction was terminated by adding an acidic stop solution. Each well’s optical density (OD) was measured using a microplate reader set to 450 nm within the specified time limit.

### 2.7. Western Blot Analysis

The levels of inflammatory markers NF-κB and interleukin-6 (IL-6) and hypoxia marker hypoxia-induced factor-1α (HIF-1α) were measured with Western blot. The markers were assessed in the cuff-induced LI group 3 h after IRI and compared with the sham group. Hindlimb tissue was homogenized using homogenizer (TissueLyser LT, Qiagen, Hilden, Germany) in homogenization buffer (HB) (pH 7.4) containing 0.25 M sucrose, 0.1 M monobasic KH_2_PO_4_, 0.1 M dibasic K_2_HPO_4_, 0.5 M EDTA, 0.8 mM DTT with addition of protease (#1862209, Fisher, Hampton, NH, USA) and phosphatase inhibitors (P0044, Sigma, St. Louis, MO, USA). The supernatant was transferred to a clean tube after centrifugation (12,000× *g* for 15 min at 4 °C). Protein concentration was measured by Bio-Rad protein assay (#5000006, Bio-Rad, Hercules, CA, USA). Samples were heated at 95 °C for 10 min in 4× SDS sample buffer containing 5% of β-mercaptoethanol, run on 4–15% polyacrylamide gel (30 ug total protein/lane), transferred to PVDF membranes, blocked in TBS (Bio-Rad, # 1706435) + Tween-20 (TBST)/5% nonfat dry milk, and probed with the primary antibody against NF-κB (ab-16502, Abcam, Cambridge, UK), IL-6 (21865-1-AP) and HIF-1α (sc-13515) diluted in TBST/3% BSA overnight (O/N) at 4 °C. After washing in TBST and incubation with the secondary antibody, diluted in 5% milk/TBST for 1 h at room temperature, proteins were detected by enhanced chemiluminescence (ECL, Pierce). Equal amounts of protein were loaded per lane as determined by a Bio-Rad protein assay and verified by blotting with housekeeping protein GAPDH (sc-32233, Santa Cruz, CA, USA). Antibodies were selected for their monospecificity and recognized as a single band of predicted molecular weight. The Image Lab software (Version 6.1, #12012931, Bio-Rad) was used for blot quantification.

### 2.8. Histology Evaluated with H&E Staining

The gastrocnemius skeletal muscle tissue was harvested and fixed in a 4% paraformaldehyde solution. Fixed tissues were embedded in paraffin, and 4 µm hindlimb tissue slices were cut and stained with hematoxylin and eosin (H&E). The degree of muscle injury for deficiency in muscle fibers, myofibrillar sparsity, necrosis, and centralized nuclei was quantified from the percentage of whole areas (<25%, 25–50%, 50–80%, >80%) [19]. The fiber area and size were calculated with Image J V2 software (National Institutes of Health, Bethesda, MD, USA). All morphometric analyses were performed blindly. Five random visual fields from each specimen were selected, photographed under a microscope (200×, Olympus BX53, Waltham, MA, USA), and analyzed using Fiji/ImageJ for statistical analysis.

### 2.9. Statistics

Experimental values are presented as mean ± SEM unless otherwise indicated in the figure legends. Statistical analysis was performed using Prism 10 (GraphPad Software; La Jolla, CA, USA) or build-in software as stated in the method. Statistical tests for each dataset are specified in the figure legends, where statistical significance is defined as *p* < 0.05. Comparisons of the creatine kinase, grip strength, and Tarlov score at 24 h were performed by one-way or two-way analysis of variance (ANOVA). Comparisons of the datasets of histology and Western blot were performed using Student’s *t* test.

## 3. Results

### 3.1. Creatine Kinase Measurements

To assess the limb skeletal muscle injury, CK levels were measured in plasma samples collected from the heart 24 h after surgery. The CK level in the cuff-induced LI group was 3.95 ± 0.18 ng/mL, which was almost two times higher compared with the sham (2.42 ± 0.4 ng/mL) and other-surgical-methods-induced LI, which were 2.39 ± 0.31, 2.40 ± 0.07, and 2.48 ± 0.41 ng/mL for the aortic, iliac, and femoral artery clamping groups, respectively (Figure 1). There were no significant differences in the CK levels among aortic, iliac, and femoral artery clamping groups.

### 3.2. Grip Strength Measurements

To measure the hindlimb muscular function, a grip strength meter was used to measure the grip strength of the left hindlimbs of the rats 24 h or up to 7 days after surgery. In the cuff-induced LI group, the grip strength of the left hindlimbs was almost lost, which was <10 units of gram force (gf). The grip strengths in all other IRI groups were at a similar level to the sham group, which were 155.62 ± 23.87 gf in the aortic, 216.15 ± 13.90 gf in the iliac, and 165.12 ± 14.62 gf in the femoral clamping group, while it was 191.13 ± 11.98 gf in the sham group (Figure 2A), indicating unaffected muscular function by ligation of aortic, iliac, or femoral. Therefore, we evaluated hindlimb function recovery in the cuff-induced IRI group by monitoring grip strength for up to 7 days and compared it to the sham group. Figure 2B shows a significant impairment in hindlimb function in the cuff group, with gradual recovery observed within 7 days based on grip strength measurements.

### 3.3. Modified Tarlov Scale Scoring Measurements

Motor functions of left hindlimbs were evaluated with a modified Tarlov scale method at 24 h and 7 days after surgery in all groups of rats. Based on a scoring system of 0–6, the cuff-induced LI group exhibited impaired motor function with an average score of 4, which gradually improved during the monitor period (Figure 3A,B). However, it did not achieve a full recovery within 7 days (Figure 3C). All other groups of rats with IRI returned to normal limb function with a score of 6, which is the same as the sham group.

### 3.4. Western Blot Analysis of Limb Samples Isolated from SD Rats

NF-κB and IL-6, inflammatory markers, and HIF-1α, a hypoxia marker, were measured in the sham and cuff-induced LI group since this was the only group of rats that exhibited impaired functional recovery. Hindlimb samples were collected 3 h after IRI. Compared with the sham group (animals that underwent the same surgical procedures but did not receive any interventions such as clamping to block the blood supply). No significant changes were observed in the expression of IL-6 or HIF-1α between control and LI groups. LI induced over a 2-fold increase in NF-κB protein levels in the hindlimbs of SD rats (0.34 ± 0.03 in sham control vs. 0.71 ± 0.05 in LI group) (Figure 4).

### 3.5. Histology

To evaluate histological muscle injury, the gastrocnemius skeletal muscle tissues from the left (ischemia) and right (control) hindlimb was isolated, fixed, and stained with hematoxylin and eosin. Following IR for 24 h, the control hindlimb tissue had conserved muscle integrity and normal peripheral myofiber nuclei while the cuff-LI hindlimb muscles exhibited noticeable morphological alterations, including pale myofibers with centralized nuclei (yellow arrows), necrotic tissue (black arrows), expended interstitial space, and inflammatory infiltrate (blue dots) (Figure 5A,B). Measurement of the individual muscle fiber area revealed highly disorganized myofibrils of varying diameters with rounded outlines in the cuff-LI group, while the sham group showed a more uniform fiber size compared to the cuff-LI group, despite there being no significant differences in average fiber size (Figure 5C,D).

## 4. Discussion and Conclusions

In the present study, we compared different methods for inducing L-IRI and evaluated the corresponding hindlimb functional recovery in SD rats. SD rats were chosen due to their similarities to humans, including having the same muscle groups in the lower limbs, similar tissue tolerance to ischemia, and comparable biochemical and histopathological responses. SD rats are also the most widely used animal model in the study of LL-IRI [20]. In the study, we compared non-invasive and invasive methods for inducing L-IRIs. Invasive techniques involved ligating the aorta, iliac artery, or femoral artery in different animal groups, while non-invasive techniques employed the use of a pneumatic cuff. We found that only the animals with cuff-induced L-IRI exhibited impaired limb functional recovery. Therefore, pneumatic-cuff-induced L-IRI is an ideal model to study limb functional recovery. However, methods with artery ligation may be good animal models for evaluation of the microcirculation alterations after L-IRI.

The pneumatic cuff used in the study was a customized vascular cuff applied as a tourniquet. The cuff was only 1 cm in width, as we wanted to ensure that IRI largely induced the function damage and that no significant muscle injuries were induced by just the compression of the cuff itself. The limb functions were evaluated by grip strength and Tarlov scale sores. We observed a loss of grip strength within the first 24 h following the L-IRI induced by the tourniquet. The limb functions gradually recovered, but they were still about 25–35% lower than those of the control group at 7 days post-L-IRI. However, all the animals in artery ligation groups showed full recovery in hindlimb function assessed by grip strength and Tarlov scores within 24 h after L-IRI. One of our study’s objectives was to develop and optimize a mechanical system that mimics the clinical gold standard, in this case, the pneumatic cuff. This system allows us to induce ischemia with minimal pressure, thereby reducing the damage caused by compression [20].

Many studies have used rodent models to induce L-IRI through acute or chronic occlusion of the femoral, iliac, or infrarenal arteries, as well as through tourniquet application. These studies primarily focused on hindlimb arterial function during or after limb ischemia but did not fully assess overall limb function [11,12,19,21,22,23]. Investigators from the University of Nebraska compared lower L-IRI induced by tourniquet application and infrarenal aortic clamping in rats [11]. They found that tourniquet application led to severe lower limb ischemia, whereas infrarenal aortic ligation did not. Similarly, L-IRI induced by tourniquet application was compared with femoral artery ligation in mouse models [12]. Consistent with our findings, these studies showed that tourniquet application induced more severe IRI than femoral artery ligation. Moreover, they observed higher proinflammatory cytokine levels in the tourniquet application group compared to the femoral artery ligation group, which aligns with our results. While these studies did not fully assess overall limb function, the observed limb injury patterns are consistent with our study, suggesting that tourniquet-induced L-IRI impairs limb functional recovery [20].

Animal models of hindlimb ischemia have been developed in rabbits, pigs, rats, and mice, with many laboratories working to create an ideal model for L-IRI [11,12,19,21,22,23]. These models have provided valuable and translational insights for the clinical treatment of L-IRI. However, it is important to recognize that, as with many other diseases, animal models cannot fully replicate the clinical conditions experienced by patients. For example, the L-IRI procedure is predominately performed in young and healthy animals, which does not reflect the clinical scenario that patients with L-IRI typically fall, usually old age, with co-morbidities like diabetes, hypertension, or hypercholesterolemia [3]. Creatine kinase (CK) is found in skeletal muscle and other tissues like the heart and brain [14]. Due to its high sensitivity, any damage to these organs can cause an increase in plasma CK levels [24]. Consequently, it is crucial to carefully distinguish the reasons for elevated CK in patients in clinical settings. However, in this study, we used normal healthy SD rats that did not have any apparent diseases or injuries in organs other than the muscles. Hence, in this context, CK can be regarded as a sensitive and specific marker for skeletal muscle injury [25].

L-IRI occurs when blood flow to a limb is temporarily restricted (ischemia) and then restored (reperfusion). This event triggers a series of biochemical and cellular responses, leading to the expression of inflammatory and hypoxia-associated proteins. During ischemia, the lack of oxygen and nutrients causes tissue hypoxia, stabilizing and activating hypoxia-inducible factors like HIF-1α, which upregulate genes involved in angiogenesis and metabolic adaptation. Reperfusion introduces a sudden influx of oxygen, generating reactive oxygen species (ROS) that exacerbate tissue damage and initiate an inflammatory response. The upregulation of master regulator proteins like NF-κB and proinflammatory cytokines such as IL-6 drives this inflammation.

In this study, the animal group subjected to cuff-induced L-IRI showed a significant increase in NF-κB expression, supporting our hypothesis. However, IL-6 expression was unexpectedly low and not significantly different in the cuff-induced L-IRI group. Interestingly, HIF-1α, a marker of cellular adaptation to hypoxia, was similarly expressed in both groups, indicating a need for further research to elucidate the molecular pathways involved in cuff-induced L-IRI. The goal of this study was achieved, demonstrating that the cuff method induces L-IRI, as evidenced by elevated plasma CK levels 24 h post-injury and increased expression of stress-related master regulator proteins.

In addition, the responses to artery ligation are strain specific. It has been reported [12] that in response to femoral artery ligation, BALB/c mice exhibited severe impairment in limb arterial flow, but C57BL/6 mice did not show significant changes in either the blood flow or muscle contraction, which agreed with our observation that ligation of femoral artery did not induce impaired limb function in C57BL/6 mice. Therefore, different animal models have different characterizations. Primarily based on the goal and scope of the research interest, animal models should be carefully selected for each project.

In conclusion, we induced L-IRI in SD rats by non-invasive tourniquet and invasive artery ligations and evaluated the functional recovery of the hindlimb. Tourniquet-cuff-induced L-IRI exhibited impaired limb functional recovery in the rats, while artery ligations did not. The invasive procedures resulted in minimal skeletal muscle injury, as evidenced by the absence of significant increases in CK levels shown in Figure 1. Limb functional recovery, assessed using Tarlov’s scale, indicated full recovery within 24 h following L-IRI induced by artery ligations (Figure 3A). Additionally, grip strength measurements (Figure 2A) indicated nearly complete recovery. Therefore, the non-invasive tourniquet method can be ideal for studying functional limb recovery after L-IRI in rat models.

## Figures and Tables

**Figure 1 biomedicines-12-02079-f001:**
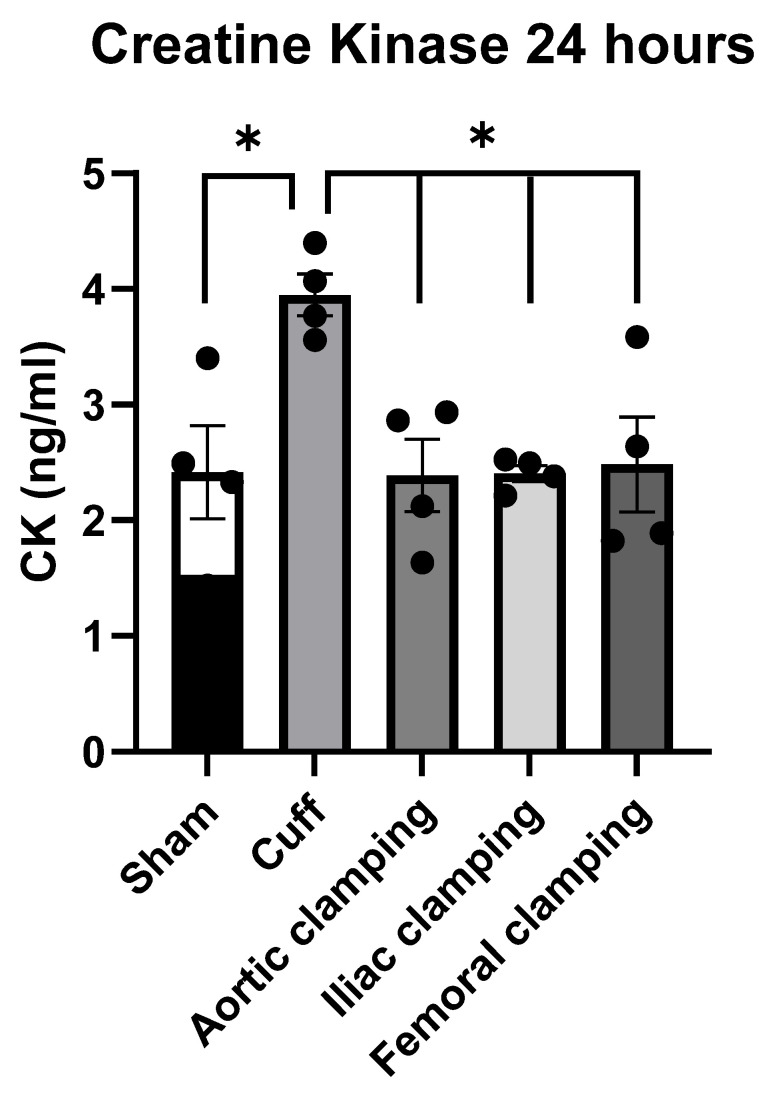
Creatine kinase (CK) concentrations. Plasma samples were collected from rats with L-IRI induced by the cuff or clamping of the aortic, iliac, or femoral artery (*n* = 4 rats for each group) 24 h after surgery and compared with the sham group. Statistical analysis was performed with one-way ANOVA; * *p* < 0.05 was considered significant.

**Figure 2 biomedicines-12-02079-f002:**
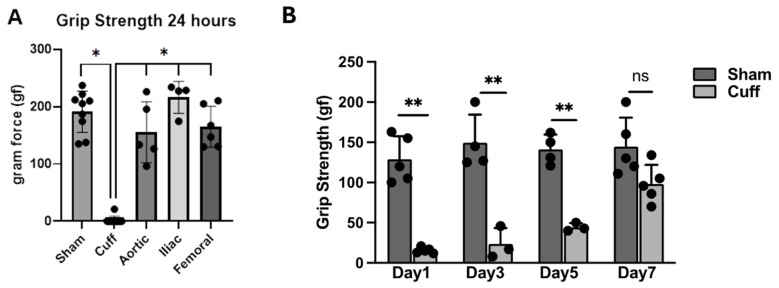
Measurement of grip strength of rats from different experimental groups. The grip strength of the left hindlimbs of the rats was measured with a grip strength meter (Ugo Basile 47,200 Grip Strength Meter) 24 h (**A**) for all groups and up to 7 days (**B**) after surgery for the cuff group. The ischemia-reperfusion injury of left hindlimbs was induced with cuff (*n* = 4) or clamping of the aortic (*n* = 5), iliac (*n* = 5), or femoral (*n* = 4) artery and compared with the sham group (*n* = 9). One-way ANOVA and two-way ANOVA followed by Sidak multiple comparison was performed to compare the grip strengths among different groups at 24 h after IRI and between the sham and cuff groups for up to 7 days, respectively (* *p* < 0.05, ** *p* < 0.01, ns, not significant).

**Figure 3 biomedicines-12-02079-f003:**
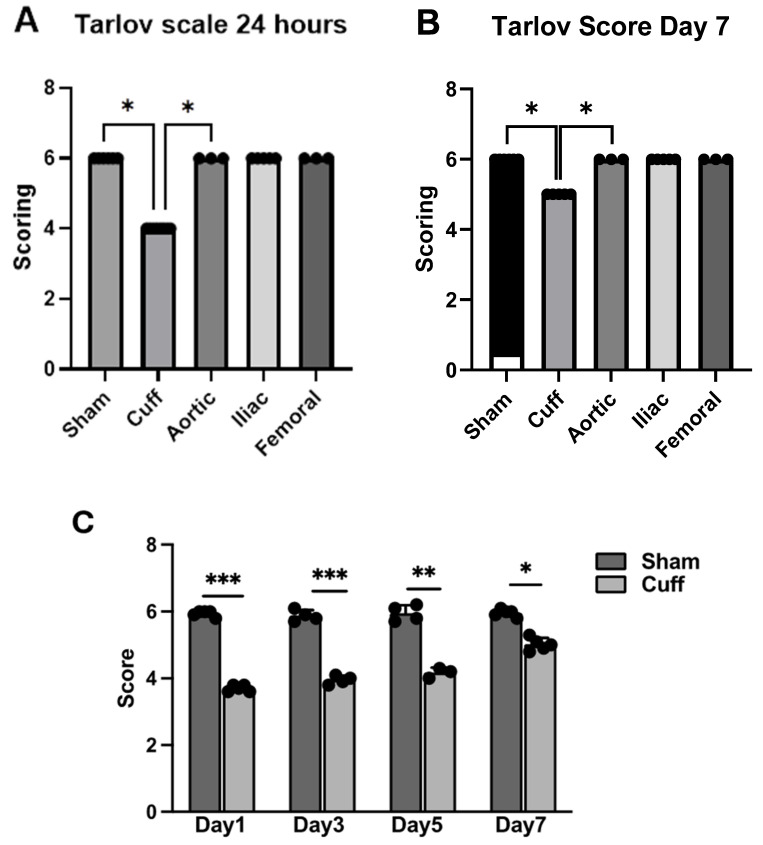
Evaluation of motor functions with a modified Tarlov scale. Motor functions of the hindlimbs were evaluated with a modified Tarlov scoring scale. The position, motion against gravity, gait, and walking ability of the animal in a transparent 1 × 2 ft bucket was observed and graded in all groups of rats (sham, cuff, aortic, iliac, femoral-clamping-induced ischemia, *n* = 4 rats/group) 24 h (**A**) and 7 days (**B**) after surgery. The recovery from L-IRI was assessed in the experimental cohort, referred to as the cuff group, over seven days (**C**). One-way ANOVA and two-way ANOVA followed by Sidak multiple comparison was performed to compare the Tarlov scales among different groups at 24 h after IRI and between the sham and cuff groups for up to 7 days, respectively. (* *p* < 0.05, ** *p* < 0.01 and *** *p* < 0.001).

**Figure 4 biomedicines-12-02079-f004:**
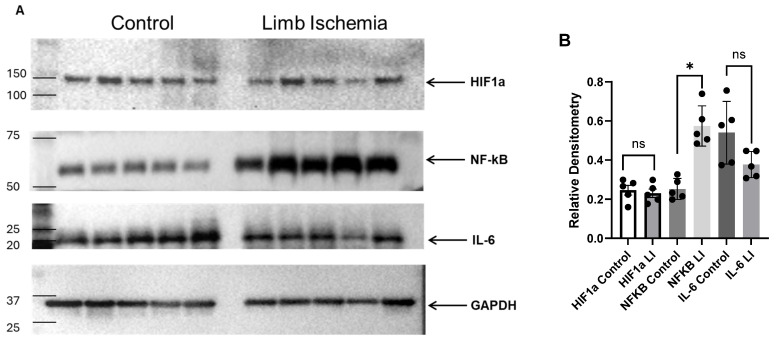
(**A**) Western blot analysis of the expression of inflammatory factors in the hindlimb tissue homogenates from SD rats after 3 h of limb ischemia compared with sham controls. The limb ischemia samples (*n* = 5) exhibited significantly higher NF-κB expression than the sham control group. The GraphPad Prism 10 software was used as a statistical analytical tool with an unpaired *t*-test. (**B**) Densitometry data: The relative expression of a target protein was calculated using the ChemiDoc MP Imager and Image Lab V 6.1 software. GAPDH was used as a loading control (* *p* < 0.05; ns, not significant).

**Figure 5 biomedicines-12-02079-f005:**
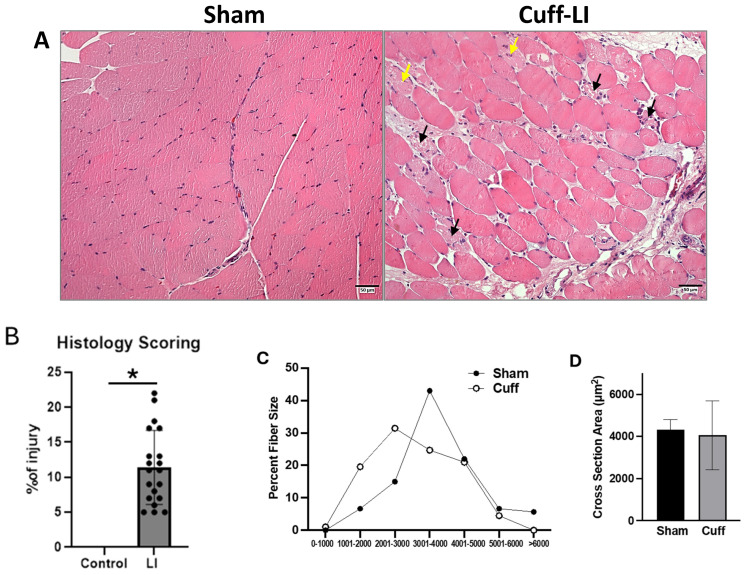
Evaluation of muscle injury with H&E staining. (**A**) Representative images of hematoxylin and eosin (H&E) staining of hindlimb muscle tissue sections from control and LI-cuff-ischemia-induced SD rat; (**B**) histological scoring was performed using the histology scoring system described in the methods; (**C**) size distribution of muscle fiber diameter; (**D**) average muscle fiber cross-section area. Student’s 1-tailed *t* test was applied (*n* = 3 mice, *n* = 140 fibers per mouse) (* *p* < 0.05).

**Table 1 biomedicines-12-02079-t001:** Modified Tarlov scale.

Score	Hindlimb Motor Function
0	no voluntary movement
1	barely perceptible movement
2	frequent movement of hindlimbs, no weight support
3	alternate stepping or propulsive movement, no weight support
4	hindlimbs can support weight
5	ambulation with mild deficit
6	normal ambulation

## Data Availability

All the data and materials supporting the findings of this study are available within the article. Further enquiries can be directed to the corresponding author.

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
