# Peer review of "Comparison of Different Animal Models in Hindlimb Functional Recovery after Acute Limb Ischemia-Reperfusion Injury"

_biomedicines, 2024, doi:10.3390/biomedicines12092079_

Round 1

Reviewer 1 Report (Previous Reviewer 1)

Comments and Suggestions for Authors

Thank you so much for submitting the revised version and addressing my suggestions.

Reviewer 2 Report (Previous Reviewer 3)

Comments and Suggestions for Authors

The authors answered to my 1st round revision clearly and modified the article according to them. The paper is suitable for publication.

Comments on the Quality of English Language

some minor changes could improve the quality

This manuscript is a resubmission of an earlier submission. The following is a list of the peer review reports and author responses from that submission.

Round 1

Reviewer 1 Report

Comments and Suggestions for Authors

Please view the report in the attached file.

Reviewer 2 Report

Comments and Suggestions for Authors

The manuscript provides a thorough comparison of non-invasive and invasive methods for inducing limb ischemia-reperfusion injury (L-IRI) in rats, which is valuable for researchers looking to choose an appropriate model for their studies. The methods section is well-detailed, allowing for reproducibility. The description of the pneumatic cuff method and the various invasive ligation techniques ensures that other researchers can replicate the study.

However, the following comments must be addressed:

Limited Scope of Biochemical Analysis: The biochemical analysis is limited to NF-κB and creatine kinase levels. Additional inflammation and muscle damage markers could provide a more comprehensive understanding of the molecular changes occurring in L-IRI.

Short Duration of Functional Assessment: Functional recovery was only assessed up to 7 days post-ischemia. Longer-term studies could provide insights into the chronic effects and recovery patterns following L-IRI.

Lack of Alternative Models: The study only uses Sprague Dawley rats. Including other strains or species could provide insights into the variability and robustness of the findings across different genetic backgrounds.

Potential for Bias in Functional Assessment: Although the grip strength and Tarlov scale assessments were conducted blindly, subjective biases could still influence the evaluation. Objective measures, such as automated gait analysis, could supplement these assessments.

Limited Clinical Translation Discussion: The discussion section could be strengthened by elaborating on how the findings translate to clinical scenarios, particularly the limitations and potential for using these models to study human L-IRI.

Histological Analysis Not Comprehensive: The histological evaluation focuses on general muscle injury but does not delve deeply into specific types of cellular damage or repair processes. Additional staining techniques and quantitative analyses could enhance this aspect.

Suggestions for Improvement

Expand Biochemical Analyses: A more comprehensive range of biochemical markers (e.g., cytokines, oxidative stress markers) would provide a more comprehensive picture of the inflammatory and injury processes.

Longer-Term Studies: Extending the duration of functional assessments beyond seven days would help understand the long-term impacts and recovery following L-IRI.

Diverse Animal Models: Incorporating different rat strains or other animal models could help generalise the findings and understand strain-specific responses.

Advanced Functional Assessments: Utilizing advanced techniques such as motion capture systems for gait analysis or electromyography for muscle function could provide more objective and detailed functional assessments.

Enhanced Histological Evaluation: Implementing additional histological techniques, such as immunohistochemistry for specific cell types or damage markers, could provide deeper insights into the tissue-level effects of L-IRI.

Comments on the Quality of English Language

Minor editing of English language required

Reviewer 3 Report

Comments and Suggestions for Authors

I’ve carefully revised this interesting manuscript, which needs just some revisions before to be considered suitable for publication.

These points should be amended to definitively improve the article:

-          In MM, please be clearer and explain for each test why sometimes you compared the five groups and in other only sham and cuff. This point is clear only for western blot (lines 181-183).

-          Please provide more details about the scoring system used to study muscle injuries by histology. In particular, please describe clearly what did you define as muscle injury, which lesion and its intensity. The noun “injury” is to imprecise and large.

-          Add in MM the part about Creatine kinase, which is present in the results without any mention before.

Comments on the Quality of English Language

Minor editing of English language required

Reviewer 4 Report

Comments and Suggestions for Authors

The paper presents a comparison of several methods used for hindlimb functional recovery after limb ischemia-reperfusion injury. The subject addressed in this paper is indeed very significant for medical and biomedical research fields, and thus, the paper has the potential to make valuable contributions to its field.

However, there are some changes that could be made to properly disseminate the findings of the study:

  1. Please define the aim of the paper in the abstract.
  2. Consider moving parts where other studies on this topic have been mentioned from the discussion section to the introduction section to provide the context that supports the necessity of your study.
  3. I recommend that the authors add a), b), etc. notations in the descriptions of the figures where needed. Also, ensure that all figures are mentioned in the text where they are explained.
  4. I suggest that the authors rename section 4 to “Discussion and Conclusion” or split the section into Section 4: Discussion and Section 5: Conclusion, moving the last paragraph of section 4 to section 5.
  5. Ensure that all acronyms are explained at their first appearance in the text.